# Sparse Support Recovery with Non-smooth Loss Functions

**Kévin Degraux**
ISPGroup/ICTEAM, FNRS
Université catholique de Louvain
Louvain-la-Neuve, Belgium 1348
`kevin.degraux@uclouvain.be`

**Gabriel Peyré**
CNRS, DMA
École Normale Supérieure
Paris, France 75775
`gabriel.peyre@ens.fr`

**Jalal M. Fadili**
Normandie Univ, ENSICAEN,
CNRS, GREYC,
Caen, France 14050
`Jalal.Fadili@ensicaen.fr`

**Laurent Jacques**
ISPGroup/ICTEAM, FNRS
Université catholique de Louvain
Louvain-la-Neuve, Belgium 1348
`laurent.jacques@uclouvain.be`

## Abstract

In this paper, we study the support recovery guarantees of underdetermined sparse regression using the $\ell_1$-norm as a regularizer and a non-smooth loss function for data fidelity. More precisely, we focus in detail on the cases of $\ell_1$ and $\ell_\infty$ losses, and contrast them with the usual $\ell_2$ loss. While these losses are routinely used to account for either sparse ($\ell_1$ loss) or uniform ($\ell_\infty$ loss) noise models, a theoretical analysis of their performance is still lacking. In this article, we extend the existing theory from the smooth $\ell_2$ case to these non-smooth cases. We derive a sharp condition which ensures that the support of the vector to recover is stable to small additive noise in the observations, as long as the loss constraint size is tuned proportionally to the noise level. A distinctive feature of our theory is that it also explains what happens when the support is unstable. While the support is not stable anymore, we identify an "extended support" and show that this extended support is stable to small additive noise. To exemplify the usefulness of our theory, we give a detailed numerical analysis of the support stability/instability of compressed sensing recovery with these different losses. This highlights different parameter regimes, ranging from total support stability to progressively increasing support instability.

## 1 Introduction

### 1.1 Sparse Regularization

This paper studies sparse linear regression problems of the form
$$y = \Phi x_0 + w,$$
where $x_0 \in \mathbb{R}^n$ is the unknown vector to estimate, supposed to be non-zero and sparse, $w \in \mathbb{R}^m$ is some additive noise and the design matrix $\Phi^{m \times n}$ is in general rank deficient corresponding to a noisy underdetermined linear system of equations, *i.e.*, typically in the high-dimensional regime where $m \ll n$. This can also be understood as an inverse problem in imaging sciences, a particular instance of which being the compressed sensing problem [3], where the matrix $\Phi$ is drawn from some appropriate random matrix ensemble.

In order to recover a sparse vector $x_0$, a popular regularization is the $\ell_1$-norm, in which case we consider the following constrained sparsity-promoting optimization problem
$$\min_{x \in \mathbb{R}^n} \left\{ \|x\|_1 \text{ s.t. } \|\Phi x - y\|_\alpha \leqslant \tau \right\}, \qquad (\mathcal{P}_\alpha^\tau(y))$$

where for $\alpha \in [1, +\infty]$, $\|u\|_\alpha \overset{\text{def.}}{=} \left( \sum_i |u_i|^\alpha \right)^{1/\alpha}$ denotes the $\ell_\alpha$-norm, and the constraint size $\tau \geqslant 0$ should be adapted to the noise level. To avoid trivialities, through the paper, we assume that problem $(\mathcal{P}_\alpha^\tau(y))$ is feasible, which is of course the case if $\tau \geqslant \|w\|_\alpha$. In the special situation where there is no noise, *i.e.*, $w = 0$, it makes sense to consider $\tau = 0$ and solve the so-called Lasso [14] or Basis-Pursuit problem [4], which is independent of $\alpha$, and reads

$$\min_x \ \{ \|x\|_1 \ \text{s.t.} \ \Phi x = \Phi x_0 \} . \qquad (\mathcal{P}^0(\Phi x_0))$$

The case $\alpha = 2$ corresponds to the usual $\ell_2$ loss function, which entails a smooth constraint set, and has been studied in depth in the literature (see Section 1.6 for an overview). In contrast, the cases $\alpha \in \{1, +\infty\}$ correspond to very different setups, where the loss function $\| \cdot \|_\alpha$ is polyhedral and non-smooth. They are expected to lead to significantly different estimation results and require to develop novel theoretical results, which is the focus of this paper. The case $\alpha = 1$ corresponds to a "robust" loss function, and is important to cope with impulse noise or outliers contaminating the data (see for instance [11, 13, 9]). At the extreme opposite, the case $\alpha = +\infty$ is typically used to handle uniform noise such as in quantization (see for instance [10]). This paper studies the stability of the support $\mathrm{supp}(x_\tau)$ of minimizers $x_\tau$ of $(\mathcal{P}_\alpha^\tau(y))$. In particular, we provide a sharp analysis for the polyhedral cases $\alpha \in \{1, +\infty\}$ that allows one to control the deviation of $\mathrm{supp}(x_\tau)$ from $\mathrm{supp}(x_0)$ if $\|w\|_\alpha$ is not too large and $\tau$ is chosen proportionally to $\|w\|_\alpha$. The general case is studied numerically in a compressed sensing experiment where we compare $\mathrm{supp}(x_\tau)$ and $\mathrm{supp}(x_0)$ for $\alpha \in [1, +\infty]$.

## 1.2 Notations.

The support of $x_0$ is noted $I \overset{\text{def.}}{=} \mathrm{supp}(x_0)$ where $\mathrm{supp}(u) \overset{\text{def.}}{=} \{i \mid u_i \neq 0\}$. The saturation support of a vector is defined as $\mathrm{sat}(u) \overset{\text{def.}}{=} \{i \mid |u_i| = \|u\|_\infty\}$. The sub-differential of a convex function $f$ is denoted $\partial f$. The subspace parallel to a nonempty convex set $\mathcal{C}$ is $\mathrm{par}(\mathcal{C}) \overset{\text{def.}}{=} \mathbb{R}(\mathcal{C} - \mathcal{C})$. $A^*$ is the transpose of a matrix $A$ and $A^+$ is the Moore-Penrose pseudo-inverse of A. Id is the identity matrix and $\delta_i$ the canonical vector of index $i$. For a subspace $V \subset \mathbb{R}^n$, $P_V$ is the orthogonal projector onto $V$. For sets of indices $S$ and $I$, we denote $\Phi_{S,I}$ the submatrix of $\Phi$ restricted to the rows indexed by $S$ and the columns indexed by $I$. When all rows or all columns are kept, a dot replaces the corresponding index set (*e.g.*, $\Phi_{\cdot,I}$). We denote $\Phi_{S,I}^* \overset{\text{def.}}{=} (\Phi_{S,I})^*$, i.e. the transposition is applied after the restriction.

## 1.3 Dual Certificates

Before diving into our theoretical contributions, we first give important definitions. Let $\mathcal{D}_{x_0}$ be the set of *dual certificates* (see, *e.g.*, [17]) defined by

$$\mathcal{D}_{x_0} \overset{\text{def.}}{=} \{p \in \mathbb{R}^m \mid \Phi^* p \in \partial \|x_0\|_1\} = \{p \in \mathbb{R}^m \mid \Phi_{\cdot,I}^* p = \mathrm{sign}(x_{0,I}), \|\Phi^* p\|_\infty \leqslant 1\}. \quad (1)$$

The first order optimality condition (see, *e.g.*, [12]) states that $x_0$ is a solution of $(\mathcal{P}^0(\Phi x_0))$ if and only if $\mathcal{D}_{x_0} \neq \emptyset$. Assuming this is the case, our main theoretical finding (Theorem 1) states that the stability (and instability) of the support of $x_0$ is characterized by the following specific subset of certificates

$$p_\beta \in \underset{p \in \mathcal{D}_{x_0}}{\mathrm{Argmin}} \ \|p\|_\beta \quad \text{where} \quad \tfrac{1}{\alpha} + \tfrac{1}{\beta} = 1. \qquad (2)$$

We call such a certificate $p_\beta$ a *minimum norm certificate*. Note that for $1 < \alpha < +\infty$, this $p_\beta$ is actually unique but that for $\alpha \in \{1, \infty\}$ it might not be the case.

Associated to such a minimal norm certificate, we define the *extended support* as

$$J \overset{\text{def.}}{=} \mathrm{sat}(\Phi^* p_\beta) = \{i \in \{1, \ldots, n\} \mid |(\Phi^* p_\beta)_i| = 1\}. \qquad (3)$$

When the certificate $p_\beta$ from which $J$ is computed is unclear from the context, we write it explicitly as an index $J_{p_\beta}$. Note that, from the definition of $\mathcal{D}_{x_0}$, one always has $I \subseteq J$. Intuitively, $J$ indicates the set of indexes that will be activated in the signal estimate when a small noise $w$ is added to the observation, and thus the situation when $I = J$ corresponds to the case where the support of $x_0$ is stable.

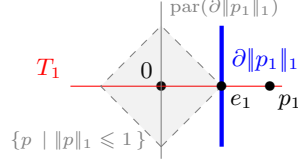

Fig. 1: *Model tangent subspace $T_\beta$ in $\mathbb{R}^2$ for $(\alpha, \beta) = (\infty, 1)$.*

### 1.4 Lagrange multipliers and restricted injectivity conditions

In the case of noiseless observations ($w = 0$) and when $\tau > 0$, the following general lemma whose proof can be found in Section 2 associate to a given dual certificate $p_\beta$ an explicit solution of $(\mathcal{P}_\alpha^\tau(\Phi x_0))$. This formula depends on a so-called *Lagrange multiplier vector* $v_\beta \in \mathbb{R}^n$, which will be instrumental to state our main contribution (Theorem 1). Note that this lemma is valid for any $\alpha \in [1, \infty]$. Even though this goes beyond the scope of our main result, one can use the same lemma for an arbitrary $\ell_\alpha$-norm for $\alpha \in [1, \infty]$ (see Section 3) or for even more general loss functions.

**Lemma 1** (Noiseless solution). *We assume that $x_0$ is identifiable, i.e. it is a solution to $(\mathcal{P}^0(\Phi x_0))$, and consider $\tau > 0$. Then there exists a $v_\beta \in \mathbb{R}^n$ supported on $J$ such that*

$$\Phi_{\cdot, J} v_{\beta, J} \in \partial \|p_\beta\|_\beta \quad and \quad -\operatorname{sign}(v_{\beta, \tilde{j}}) = \Phi_{\cdot, \tilde{j}}^* p_\beta$$

*where we denoted $\tilde{J} \overset{\text{def.}}{=} J \setminus I$. If $\tau$ is such that $0 < \tau < \frac{\underline{x}}{\|v_{\beta, I}\|_\infty}$, with $\underline{x} = \min_{i \in I} |x_{0, I}|$, then a solution $\bar{x}_\tau$ of $(\mathcal{P}_\alpha^\tau(\Phi x_0))$ with support equal to $J$ is given by*

$$\bar{x}_{\tau, J} = x_{0, J} - \tau v_{\beta, J}.$$

*Moreover, its entries have the same sign as those of $x_0$ on its support $I$, i.e., $\operatorname{sign}(\bar{x}_{\tau, I}) = \operatorname{sign}(x_{0, I})$.*

An important question that arises is whether $v_\beta$ can be computed explicitly. For this, let us define the *model tangent subspace* $T_\beta \overset{\text{def.}}{=} \operatorname{par}(\partial \|p_\beta\|_\beta)^\perp$, *i.e.*, $T_\beta$ is the orthogonal to the subspace parallel to $\partial \|p_\beta\|_\beta$, which uniquely defines the *model vector*, $e_\beta \overset{\text{def.}}{=} P_{T_\beta} \partial \|p_\beta\|_\beta$, as shown on Figure 1 (see [17] for details). Using this notation, $v_{\beta, J}$ is uniquely defined and expressed in closed-form as

$$v_{\beta, J} = (P_{T_\beta} \Phi_{\cdot, J})^+ e_\beta \tag{4}$$

if and only if the following *restricted injectivity condition* holds

$$\operatorname{Ker}(P_{T_\beta} \Phi_{\cdot, J}) = \{0\}. \tag{INJ$_\alpha$}$$

For the special case $(\alpha, \beta) = (\infty, 1)$, the following lemma, proved in Section 2, gives easily verifiable sufficient conditions, which ensure that (INJ$_\infty$) holds. The notation $S \overset{\text{def.}}{=} \operatorname{supp}(p_1)$ is used.

**Lemma 2** (Restricted injectivity for $\alpha = \infty$). *Assume $x_0$ is identifiable and $\Phi_{S, J}$ has full rank. If*

$$s_J \notin \operatorname{Im}(\Phi_{S', J}^*) \quad \forall S' \subseteq \{1, \ldots, m\}, \quad |S'| < |J| \quad and$$

$$q_S \notin \operatorname{Im}(\Phi_{S, J'}) \quad \forall J' \subseteq \{1, \ldots, n\}, \quad |J'| < |S|,$$

*where $s_J = \Phi_{\cdot, J}^* p_1 \in \{-1, 1\}^{|J|}$, and $q_S = \operatorname{sign}(p_{1, S}) \in \{-1, 1\}^{|S|}$, then, $|S| = |J|$ and $\Phi_{S, J}$ is invertible, i.e., since $P_{T_1} \Phi_{\cdot, J} = \operatorname{Id}_{\cdot, S} \Phi_{S, J}$, (INJ$_\infty$) holds.*

*Remark* 1. If $\Phi$ is randomly drawn from a continuous distribution with i.i.d. entries, *e.g.*, Gaussian, then as soon as $x_0$ is identifiable, the conditions of Lemma 2 hold with probability 1 over the distribution of $\Phi$.

For $(\alpha, \beta) = (1, \infty)$, we define $Z \overset{\text{def.}}{=} \operatorname{sat}(p_\infty)$,

$$\Theta \overset{\text{def.}}{=} \begin{bmatrix} \operatorname{Id}_{Z^c, \cdot} \\ \operatorname{sign}(p_{\infty, Z}^*) \operatorname{Id}_{Z, \cdot} \end{bmatrix} \quad and \quad \widetilde{\Phi} \overset{\text{def.}}{=} \Theta \Phi_{\cdot, J}.$$

Following similar reasoning as in Lemma 2 and Remark 1, we can reasonably assume that $|Z^c| + 1 = |J|$ and $\widetilde{\Phi}$ is invertible. In that case, (INJ$_1$) holds as $\operatorname{Ker}(P_{T_\infty} \Phi_{\cdot, J}) = \operatorname{Ker}(\widetilde{\Phi})$. Table 1 summarizes for the three specific cases $\alpha \in \{1, 2, +\infty\}$ the quantities introduced here.

Table 1: Model tangent subspace, restricted injectivity condition and Lagrange multipliers.

| $\alpha$ | $T_\beta$ | (INJ$_\alpha$) | $(P_{T_\beta} \Phi_{\cdot, J})^+$ | $v_{\beta, J}$ |
|---|---|---|---|---|
| 2 | $\mathbb{R}^m$ | $\operatorname{Ker}(\Phi_{\cdot, J}) = \{0\}$ | $\Phi_{\cdot, J}^+$ | $\Phi_{\cdot, J}^+ \frac{p_2}{\|p_2\|_2}$ |
| $\infty$ | $\{u \mid \operatorname{supp}(u) = S\}$ | $\operatorname{Ker}(\Phi_{S, J}) = \{0\}$ | $\Phi_{S, J}^{-1} \operatorname{Id}_{S, \cdot}$ | $\Phi_{S, J}^{-1} \operatorname{sign}(p_{1, S})$ |
| 1 | $\{u \mid u_Z = \rho \operatorname{sign}(p_{\infty, Z}), \rho \in \mathbb{R}\}$ | $\operatorname{Ker}(\widetilde{\Phi}) = \{0\}$ | $\widetilde{\Phi}^{-1} \Theta$ | $\widetilde{\Phi}^{-1} \delta_{|J|}$ |

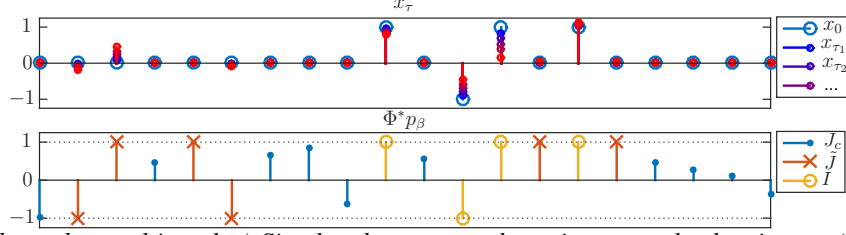

Fig. 2: *(best observed in color) Simulated compressed sensing example showing $x_\tau$ (above) for increasing values of $\tau$ and random noise $w$ respecting the hypothesis of Theorem 1 and $\Phi^* p_\beta$ (bellow) which predicts the support of $x_\tau$ when $\tau > 0$.*

## 1.5 Main result

Our main contribution is Theorem 1 below. A similar result is known to hold in the case of the smooth $\ell_2$ loss ($\alpha = 2$, see Section 1.6). Our paper extends it to the more challenging case of non-smooth losses $\alpha \in \{1, +\infty\}$. The proof for $\alpha = +\infty$ is detailed in Section 2. It is important to emphasize that the proof strategy is significantly different from the classical approach developed for $\alpha = 2$, mainly because of the lack of smoothness of the loss function. The proof for $\alpha = 1$ follows a similar structure, and due to space limitation, it can be found in the supplementary material.

**Theorem 1.** *Let $\alpha \in \{1, 2, +\infty\}$. Suppose that $x_0$ is* identifiable*, and let $p_\beta$ be a minimal norm certificate (see* (2)*) with associated extended support $J$ (see* (3)*). Suppose that the restricted injectivity condition* (INJ$_\alpha$) *is satisfied so that $v_{\beta, J}$ can be explicitly computed (see* (4)*). Then there exist constants $c_1, c_2 > 0$ depending only on $\Phi$ and $p_\beta$ such that, for any $(w, \tau)$ satisfying*

$$\|w\|_\alpha < c_1 \tau \quad and \quad \tau \leqslant c_2 \underline{x} \quad where \quad \underline{x} \stackrel{\text{def.}}{=} \min_{i \in I} |x_{0,I}|, \tag{5}$$

*a solution $x_\tau$ of $(\mathcal{P}_\alpha^\tau(\Phi x_0 + w))$ with support equal to $J$ is given by*

$$x_{\tau, J} \stackrel{\text{def.}}{=} x_{0,J} + (P_{T_\beta} \Phi_{\cdot,J})^+ w - \tau v_{\beta,J}. \tag{6}$$

This theorem shows that if the signal-to-noise ratio is large enough and $\tau$ is chosen in proportion to the noise level $\|w\|_\alpha$, then there is a solution supported exactly in the extended support $J$. Note in particular that this solution (6) has the correct sign pattern $\text{sign}(x_{\tau,I}) = \text{sign}(x_{0,I})$, but might exhibit outliers if $\tilde{J} \stackrel{\text{def.}}{=} J \backslash I \neq \emptyset$. The special case $I = J$ characterizes the exact support stability ("sparsistency"), and in the case $\alpha = 2$, the assumptions involving the dual certificate correspond to a condition often referred to as "irrepresentable condition" in the literature (see Section 1.6).

In Section 3, we propose numerical simulations to illustrate our theoretical findings on a compressed sensing (CS) scenario. Using Theorem 1, we are able to numerically assess the degree of support instability of CS recovery using $\ell_\alpha$ fidelity. As a prelude to shed light on this result, we show on Figure 2, a smaller simulated CS example for $(\alpha, \beta) = (\infty, 1)$. The parameters are $n = 20$, $m = 10$ and $|I| = 4$ and $x_0$ and $\Phi$ are generated as in the experiment of Section 3 and we use CVX/MOSEK [8, 7] at best precision to solve the optimization programs. First, we observe that $x_0$ is indeed identifiable by solving $(\mathcal{P}^0(\Phi x_0))$. Then we solve (2) to compute $p_\beta$ and predict the extended support $J$. Finally, we add uniformly distributed noise $w$ with $w_i \sim_{i.i.d.} \mathcal{U}(-\delta, \delta)$ and $\delta$ chosen appropriately to ensure that the hypotheses hold and we solve $(\mathcal{P}_\alpha^\tau(y))$. Observe that as we increase $\tau$, new non-zero entries appear in $x_\tau$ but because $w$ and $\tau$ are small enough, as predicted, we have $\text{supp}(x_\tau) = J$.

Let us now comment on the limitations of our analysis. First, this result does not trivially extend to the general case $\alpha \in [1, +\infty]$ as there is, in general, no simple closed form for $x_\tau$. A generalization would require more material and is out of the scope of this paper. Nevertheless, our simulations in Section 3 stand for arbitrary $\alpha \in [1, +\infty]$ which is why the general formulation was presented.

Second, larger noise regime, though interesting, is also out of the scope. Let us note that no other results in the literature (even for $\ell_2$) provide any insight about sparsistency in the large noise regime. In that case, we are only able to provide bounds on the distance between $x_0$ and the recovered vector but this is the subject of a forthcoming paper.

Finally our work is agnostic with respect to the noise models. Being able to distinguish between different noise models would require further analysis of the constant involved and some additional constraint on $\Phi$. However, our result is a big step towards the understanding of the solutions behavior and can be used in this analysis.

## 1.6 Relation to Prior Works

To the best of our knowledge, Theorem 1 is the first to study the support stability guarantees by minimizing the $\ell_1$-norm with non-smooth loss function, and in particular here the $\ell_1$ and $\ell_\infty$ losses. The smooth case $\alpha = 2$ is however much more studied, and in particular, the associated support stability results we state here are now well understood. Note that most of the corresponding literature studies in general the penalized form, *i.e.*, $\min_x \frac{1}{2}\|\Phi x - y\|^2 + \lambda\|x\|_1$ instead of our constrained formulation $(\mathcal{P}_\alpha^\tau(y))$. In the case $\alpha = 2$, since the loss is smooth, this distinction is minor and the proof is almost the same for both settings. However, for $\alpha \in \{1, +\infty\}$, it is crucial to study the constrained problems to be able to state our results. The support stability (also called "sparsistency", corresponding to the special case $I = J$ of our result) of $(\mathcal{P}_\alpha^\tau(y))$ in the case $\alpha = 2$ has been proved by several authors in slightly different setups. In the signal processing literature, this result can be traced back to the early work of J-J. Fuchs [6] who showed Theorem 1 when $\alpha = 2$ and $I = J$. In the statistics literature, sparsistency is also proved in [19] in the case where $\Phi$ is random, the result of support stability being then claimed with high probability. The condition that $I = J$, *i.e.*, that the minimal norm certificate $p_\beta$ (for $\alpha = \beta = 2$) is saturating only on the support, is often coined the "irrepresentable condition" in the statistics and machine learning literature. These results have been extended recently in [5] to the case where the support $I$ is not stable, i.e. $I \subsetneq J$. One could also cite [15], whose results are somewhat connected but are restricted to the $\ell_2$ loss and do not hold in our case. Note that "sparsistency"-like results have been proved for many "low-complexity" regularizers beyond the $\ell_1$-norm. Let us quote among others: the group-lasso [1], the nuclear norm [2], the total variation [16] and a very general class of "partly-smooth" regularizers [17]. Let us also point out that one of the main sources of application of these results is the analysis of the performance of compressed sensing problems, where the randomness of $\Phi$ allows to derive sharp sample complexity bounds as a function of the sparsity of $x_0$ and $n$, see for instance [18]. Let us also stress that these support recovery results are different from those obtained using tools such as the Restricted Isometry Property and alike (see for instance [3]) in many respects. For instance, the guarantees they provide are uniform (*i.e.*, they hold for any sparse enough vector $x_0$), though they usually lead to quite pessimistic worst-case bounds, and the stability is measured in $\ell_2$ sense.

## 2 Proof of Theorem 1

In this section, we prove the main result of this paper. For the sake of brevity, when part of the proof will become specific to a particular choice of $\alpha$, we will only write the details for $\alpha = \infty$. The details of the proof for $\alpha = 1$ can be found in the supplementary material.

It can be shown that the Fenchel-Rockafellar dual problem to $(\mathcal{P}_\alpha^\tau(y))$ is [12]

$$\min_{p\in\mathbb{R}^m} \ \{-\langle y, \, p\rangle + \tau\|p\|_\beta \text{ s.t. } \|\Phi^* p\|_\infty \leqslant 1\}. \qquad (\mathcal{D}_\beta^\tau(y))$$

From the corresponding (primal-dual) extremality relations, one can deduce that $(\hat{x}, \hat{p})$ is an optimal primal-dual Kuhn-Tucker pair if, and only if,

$$\Phi_{\cdot,\hat{I}}^* \hat{p} = \text{sign}(\hat{x}_{\hat{I}}) \quad \text{and} \quad \|\Phi^* \hat{p}\|_\infty \leqslant 1. \qquad (7)$$

where $\hat{I} = \text{supp}(\hat{x})$, and

$$\frac{y - \Phi\hat{x}}{\tau} \in \partial\|\hat{p}\|_\beta. \qquad (8)$$

The first relationship comes from the sub-differential of the $\ell_1$ regularization term while the second is specific to a particular choice of $\alpha$ for the $\ell_\alpha$-norm data fidelity constraint. We start by proving the Lemma 1 and Lemma 2.

**Proof of Lemma 1** Let us rewrite the problem (2) by introducing the auxiliary variable $\eta = \Phi^* p$ as

$$\min_{p,\eta} \left\{\|p\|_\beta + \iota_{B_\infty}(\eta) \mid \eta = \Phi^* p, \eta_I = \text{sign}(x_{0,I})\right\}, \qquad (9)$$

where $\iota_{B_\infty}$ is the indicator function of the unit $\ell_\infty$ ball. Define the Lagrange multipliers $v$ and $z_I$ and the associated Lagrangian function

$$\mathcal{L}(p, \eta, v, z_I) = \|p\|_\beta + \iota_{B_\infty}(\eta) + \langle v, \, \eta - \Phi^* p\rangle + \langle z_I, \, \eta_I - \text{sign}(x_{0,I})\rangle.$$

Defining $z_{I^c} = 0$, the first order optimality conditions (generalized KKT conditions) for $p$ and $\eta$ read

$$\Phi v \in \partial\|p\|_\beta \quad \text{and} \quad -v - z \in \partial\iota_{B_\infty}(\eta),$$

From the normal cone of the $B_\infty$ at $\eta$ on its boundary, the second condition is
$$-v - z \in \{u \mid u_{J^c} = 0, \text{sign}(u_J) = \eta_J\},$$
where $J = \text{sat}(\eta) = \text{sat}(\Phi^* p)$. Since $I \subseteq J$, $v$ is supported on $J$. Moreover, on $\tilde{J} = J \backslash I$, we have $-\text{sign}(v_{\tilde{J}}) = \eta_{\tilde{J}}$. As $p_\beta$ is a solution to (9), we can define a corresponding vector of Lagrange multipliers $v_\beta$ supported on $J$ such that $-\text{sign}(v_{\beta,\tilde{J}}) = \Phi^*_{\cdot,\tilde{J}} p_\beta$ and $\Phi_{\cdot,J} v_{\beta,J} \in \partial \|p_\beta\|_\beta$.

To prove the lemma, it remains to show that $\bar{x}_\tau$ is indeed a solution to $(\mathcal{P}^\tau_\alpha(y))$, i.e., it obeys (7) and (8) for some dual variable $\hat{p}$. We will show that this is the case with $\hat{p} = p_\beta$. Observe that $p_\beta \neq 0$ as otherwise, it would mean that $x_0 = 0$, which contradicts our initial assumption of non-zero $x_0$. We can then directly see that (8) is satisfied. Indeed, noting $y_0 \overset{\text{def.}}{=} \Phi x_0$, we can write
$$y_0 - \Phi_{\cdot,J} \bar{x}_{\tau,J} = \tau \Phi_{\cdot,J} v_{\beta,J} \in \tau \partial \|p_\beta\|_\beta.$$
By definition of $p_\beta$, we have $\|\Phi^* p_\beta\|_\infty \leqslant 1$. In addition, it must satisfy $\Phi^*_{\cdot,J} p_\beta = \text{sign}(\bar{x}_{\tau,J})$. Outside $I$, the condition is always satisfied since $-\text{sign}(v_{\beta,\tilde{J}}) = \Phi^*_{\cdot,\tilde{J}} p_\beta$. On $I$, we know that $\Phi^*_{\cdot,I} p_\beta = \text{sign}(x_{0,I})$. The condition on $\tau$ is thus $|x_{0,i}| > \tau |v_{\beta,i}|, \forall i \in I$, or equivalently, $\tau < \frac{x}{\|v_{\beta,I}\|_\infty}$. $\qquad\square$

**Proof of Lemma 2**  As established by Lemma 1, the existence of $p_1$ and of $v_1$ are implied by the identifiability of $x_0$. We have the following,
$$\exists p_1 \Rightarrow \exists p_S, \Phi^*_{S,J} p_S = s_J \Leftrightarrow \Phi^*_{S,J} \text{ is surjective} \Leftrightarrow |S| \geqslant |J|$$
$$\exists v_1 \Rightarrow \exists v_J, \Phi_{S,J} v_J = q_S \Leftrightarrow \Phi_{S,J} \text{ is surjective} \Leftrightarrow |J| \geqslant |S|,$$

To clarify, we detail the first line. Since $\Phi^*_{S,J}$ is full rank, $|S| \geqslant |J|$ is equivalent to surjectivity. Assume $\Phi^*_{S,J}$ is not surjective so that $|S| < |J|$, then $s_J \notin \text{Im}(\Phi^*_{S,J})$ and the over-determined system $\Phi^*_{S,J} p_S = s_J$ has no solution in $p_S$, which contradicts the existence of $p_1$. Now assume $\Phi^*_{S,J}$ is surjective, then we can take $p_S = \Phi^{*,\dagger}_{S,J} s_J$ as a solution where $\Phi^{*,\dagger}_{S,J}$ is any right-inverse of $\Phi^*_{S,J}$. This proves that $\Phi_{S,J}$ is invertible. $\qquad\square$

We are now ready to prove the main result in the particular case $\alpha = \infty$.

**Proof of Theorem 1** ($\alpha = \infty$)  Our proof consists in constructing a vector supported on $J$, obeying the implicit relationship (6) and which is indeed a solution to $(\mathcal{P}^\tau_\infty(\Phi x_0 + w))$ for an appropriate regime of the parameters $(\tau, \|w\|_\alpha)$. Note that we assume that the hypothesis of Lemma 2 on $\Phi$ holds and in particular, $\Phi_{S,J}$ is invertible. When $(\alpha, \beta) = (\infty, 1)$, the first order condition (8), which holds for any optimal primal-dual pair $(x,p)$, reads, with $S_p \overset{\text{def.}}{=} \text{supp}(p)$,
$$y_{S_p} - \Phi_{S_p,\cdot} x = \tau \text{sign}(p_{S_p}) \quad \text{and} \quad \|y - \Phi x\|_\infty \leqslant \tau. \tag{10}$$

One should then look for a candidate primal-dual pair $(\hat{x}, \hat{p})$ such that $\text{supp}(\hat{x}) = J$ and satisfying
$$y_{S_{\hat{p}}} - \Phi_{S_{\hat{p}},J} \hat{x}_J = \tau \text{sign}(\hat{p}_{S_{\hat{p}}}). \tag{11}$$

We now need to show that the first order conditions (7) and (10) hold for some $p = \hat{p}$ solution of the "perturbed" dual problem $(\mathcal{D}^\tau_1(\Phi x_0 + w))$ with $x = \hat{x}$. Actually, we will show that under the conditions of the theorem, this holds for $\hat{p} = p_1$, i.e., $p_1$ is solution of $(\mathcal{D}^\tau_1(\Phi x_0 + w))$ so that
$$\hat{x}_J = \Phi^{-1}_{S,J} y_S - \tau \Phi^{-1}_{S,J} \text{sign}(p_{1,S}) = x_{0,J} + \Phi^{-1}_{S,J} w_S - \tau v_{1,J}.$$

Let us start by proving the equality part of (7), $\Phi^*_{S,J} \hat{p}_S = \text{sign}(\hat{x}_J)$. Since $\Phi_{S,J}$ is invertible, we have $\hat{p}_S = p_{1,S}$ if and only if $\text{sign}(\hat{x}_J) = \Phi^*_{S,J} p_{1,S}$. Noting $\text{Id}_{I,J}$ the restriction from $J$ to $I$, we have
$$\text{sign}\left(x_{0,I} + \text{Id}_{I,J} \Phi^{-1}_{S,J} w_S - \tau v_{1,I}\right) = \text{sign}\left(x_{0,I}\right)$$
as soon as
$$\left|\left(\Phi^{-1}_{S,J} w_S\right)_i - \tau v_{1,i}\right| < |x_{0,I}| \quad \forall i \in I.$$
It is sufficient to require
$$\|\text{Id}_{I,J} \Phi^{-1}_{S,J} w_S - \tau v_{1,I}\|_\infty < \underline{x}$$
$$\|\Phi^{-1}_{S,J}\|_{\infty,\infty} \|w\|_\infty + \tau \|v_{1,I}\|_\infty < \underline{x},$$
with $\underline{x} = \min_{i \in I} |x_{0,I}|$. Injecting the fact that $\|w\|_\infty < c_1 \tau$ (the value of $c_1$ will be derived later), we get the condition

$$\tau \left( bc_1 + \nu \right) \leqslant \underline{x},$$

with $b = \|\Phi_{S,J}^{-1}\|_{\infty,\infty}$ and $\nu = \|v_1\|_\infty \leqslant b$. Rearranging the terms, we obtain

$$\tau \leqslant \frac{\underline{x}}{bc_1 + \nu} = c_2 \underline{x},$$

which guarantees $\operatorname{sign}(\hat{x}_I) = \operatorname{sign}(x_{0,I})$. Outside $I$, defining $\operatorname{Id}_{\tilde{J},J}$ as the restriction from $J$ to $\tilde{J}$, we must have

$$\Phi_{S,\tilde{J}}^* p_{1,S} = \operatorname{sign}\left( \operatorname{Id}_{\tilde{J},J} \Phi_{S,J}^{-1} w_S - \tau v_{1,\tilde{J}} \right).$$

From Lemma 1, we know that $-\operatorname{sign}(v_{1,\tilde{J}}) = \Phi_{S,\tilde{J}}^* p_{1,S}$, so that the condition is satisfied as soon as

$$\left| \left( \Phi_{S,J}^{-1} w_S \right)_j \right| < \tau |v_{1,j}| \quad \forall j \in \tilde{J}.$$

Noting $\underline{v} = \min_{j \in \tilde{J}} |v_{1,j}|$, we get the sufficient condition for (7),

$$\|\Phi_{S,J}^{-1} w_S\|_\infty < \tau \underline{v},$$

$$\|w\|_\infty < \tau \frac{\underline{v}}{b}. \tag{$c_1$a}$$

We can now verify (10). From (11) we see that the equality part is satisfied on $S$. Outside $S$, we have

$$y_{S^c} - \Phi_{S^c,.} \hat{x} = w_{S^c} - \Phi_{S^c,J} \Phi_{S,J}^{-1} w_S + \tau \Phi_{S^c,J} v_{1,J},$$

which must be smaller than $\tau$, *i.e.*,

$$\|w_{S^c} - \Phi_{S^c,J} \Phi_{S,J}^{-1} w_S + \tau \Phi_{S^c,J} v_{1,J}\|_\infty \leqslant \tau.$$

It is thus sufficient to have

$$(1 + \|\Phi_{S^c,J} \Phi_{S,J}^{-1}\|_{\infty,\infty}) \|w\|_\infty + \tau \mu \leqslant \tau,$$

with $\mu \overset{\text{def.}}{=} \|\Phi_{S^c,J} v_{1,J}\|_\infty$. Noting $a = \|\Phi_{S^c,J} \Phi_{S,J}^{-1}\|_{\infty,\infty}$, we get

$$\|w\|_\infty \leqslant \frac{1-\mu}{1+a} \tau. \tag{$c_1$b}$$

($c_1$a) and ($c_1$b) together give the value of $c_1$. This ensures that the inequality part of (10) is satisfied for $\hat{x}$ and with that, that $\hat{x}$ is solution to $(\mathcal{P}_\infty^\tau(\Phi x_0 + w))$ and $p_1$ solution to $(\mathcal{D}_1^\tau(\Phi x_0 + w))$, which concludes the proof. $\qquad\square$

*Remark* 2. From Lemma 1, we know that in all generality $\mu \leqslant 1$. If the inequality was saturated, it would mean that $c_1 = 0$ and no noise would be allowed. Fortunately, it is easy to prove that under a mild assumption on $\Phi$, similar to the one of Lemma 2 (which holds with probability 1 for Gaussian matrices), the inequality is strict, *i.e.*, $\mu < 1$.

## 3 Numerical experiments

In order to illustrate support stability in Lemma 1 and Theorem 1, we address numerically the problem of comparing $\operatorname{supp}(x_\tau)$ and $\operatorname{supp}(x_0)$ in a compressed sensing setting. Theorem 1 shows that $\operatorname{supp}(x_\tau)$ does not depend on $w$ (as long as it is small enough); simulations thus do not involve noise. All computations are done in Matlab, using CVX [8, 7], with the MOSEK solver at "best" precision setting to solve the convex problems. We set $n = 1000$, $m = 900$ and generate 200 times a random sensing matrix $\Phi \in \mathbb{R}^{m \times n}$ with $\Phi_{ij} \sim_{\text{i.i.d}} \mathcal{N}(0, 1)$. For each sensing matrix, we generate 60 different $k$-sparse vectors $x_0$ with support $I$ where $k \overset{\text{def.}}{=} |I|$ varies from 10 to 600. The non-zero entries of $x_0$ are randomly picked in $\{\pm 1\}$ with equal probability. Note that this choice does not impact the result because the definition of $J_{p_\beta}$ only depends on $\operatorname{sign}(x_0)$ (see (1)). It will only affect the bounds in (5). For each case, we verify that $x_0$ is identifiable and for $\alpha \in \{1, 2, \infty\}$ (which correspond to $\beta \in \{\infty, 2, 1\}$), we compute the minimum $\ell_\beta$-norm certificate $p_\beta$, solution to (2) and in particular, the support excess $\tilde{J}_{p_\beta} \overset{\text{def.}}{=} \operatorname{sat}(\Phi^* p_\beta) \backslash I$. It is important to emphasize that there is no noise in these simulations. As long as the hypotheses of the theorem are satisfied, we can predict that $\operatorname{supp}(x_\tau) = J_{p_\beta} \subset I$ without actually computing $x_\tau$, or choosing $\tau$, or generating $w$.

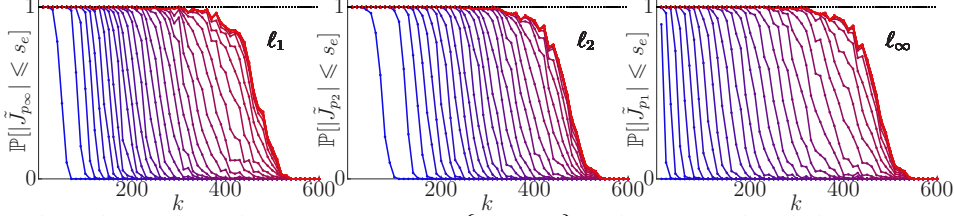

Fig. 3: *(best observed in color) Sweep over $s_e \in \{0, 10, ...\}$ of the empirical probability as a function of the sparsity $k$ that $x_0$ is identifiable and $|\tilde{J}_{p_\infty}| \leqslant s_e$ (left), $|\tilde{J}_{p_2}| \leqslant s_e$ (middle) or $|\tilde{J}_{p_1}| \leqslant s_e$ (right). The bluest corresponds to $s_e = 0$ and the redest to the maximal empirical value of $|\tilde{J}_{p_\beta}|$.*

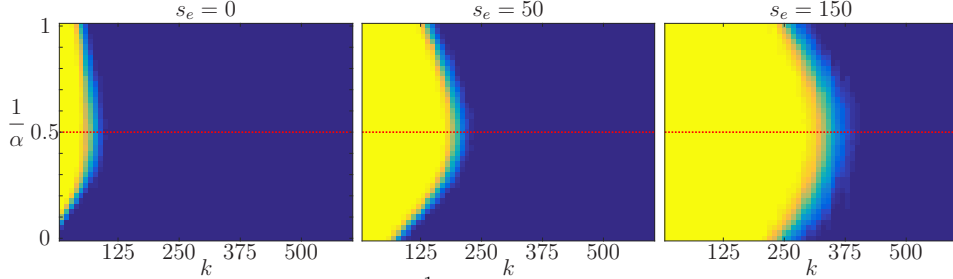

Fig. 4: *(best observed in color) Sweep over $\frac{1}{\alpha} \in [0, 1]$ of the empirical probability as a function of $k$ that $x_0$ is identifiable and $|\tilde{J}_{p_\beta}| \leqslant s_e$ for three values of $s_e$. The dotted red line indicates $\alpha = 2$.*

We define a *support excess threshold* $s_e \in \mathbb{N}$ varying from 0 to $\infty$. On Figure 3 we plot the probability that $x_0$ is identifiable *and* $|\tilde{J}_{p_\beta}|$, the cardinality of the predicted support excess, is smaller or equal to $s_e$. It is interesting to note that the probability that $|\tilde{J}_{p_1}| = 0$ (the bluest horizontal curve on the right plot) is 0, which means that even for extreme sparsity ($k = 10$) and a relatively high $m/n$ rate of 0.9, the support is never predicted as perfectly stable for $\alpha = \infty$ in this experiment. We can observe as a rule of thumb, that a support excess of $|\tilde{J}_{p_1}| \approx k$ is much more likely. In comparison, $\ell_2$ recovery provides a much more likely perfect support stability for $k$ not too large and the expected size of $\tilde{J}_{p_2}$ increases slower with $k$. Finally, we can comment that the support stability with $\ell_1$ data fidelity is in between. It is possible to recover the support perfectly but the requirement on $k$ is a bit more restrictive than with $\ell_2$ fidelity.

As previously noted, Lemma 1 and its proof remain valid for smooth loss functions such as the $\ell_\alpha$-norm when $\alpha \in (1, \infty)$. Therefore, it makes sense to compare the results with the ones obtained for $\alpha \in (1, \infty)$. On Figure 4 we display the result of the same experiment but with $1/\alpha$ as the vertical axis. To realize the figure, we compute $p_\beta$ and $\tilde{J}_{p_\beta}$ for $\beta$ corresponding to 41 equispaced values of $1/\alpha \in [0, 1]$. The probability that $|\tilde{J}_{p_\beta}| \leqslant s_e$ is represented by the color intensity. The three different plots correspond to three different values for $s_e$. On this figure, the yellow to blue transition can be interpreted as the maximal $k$ to ensure, with high probability, that $|\tilde{J}_{p_\beta}|$ does not exceeds $s_e$. It is always (for all $s_e$) further to the right at $\alpha = 2$. It means that the $\ell_2$ data fidelity constraint provides the highest support stability. Interestingly, we can observe that this maximal $k$ decreases gracefully as $\alpha$ moves away from 2 in one way or the other. Finally, as already observed on Figure 3, we see that, especially when $s_e$ is small, the $\ell_1$ loss function has a small advantage over the $\ell_\infty$ loss.

## 4 Conclusion

In this paper, we provided sharp theoretical guarantees for stable support recovery under small enough noise by $\ell_1$ minimization with non-smooth loss functions. Unlike the classical setting where the data loss is smooth, our analysis reveals the difficulties arising from non-smoothness, which necessitated a novel proof strategy. Though we focused here on the case of $\ell_\alpha$ data loss functions, for $\alpha \in \{1, 2, \infty\}$, our analysis can be extended to more general non-smooth losses, including coercive gauges. This will be our next milestone.

**Acknowledgments**

KD and LJ are funded by the Belgian F.R.S.-FNRS. JF is partly supported by Institut Universitaire de France. GP is supported by the European Research Council (ERC project SIGMA-Vision).

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
