[Supplementary Material]

# Sparse Support Recovery with Non-smooth Loss Functions: Supplementary Material

**Kévin Degraux**
ISPGroup/ICTEAM, FNRS
Université catholique de Louvain
Louvain-la-Neuve, Belgium 1348
kevin.degraux@uclouvain.be

**Gabriel Peyré**
CNRS, DMA
École Normale Supérieure
Paris, France 75775
gabriel.peyre@ens.fr

**Jalal M. Fadili**
Normandie Univ, ENSICAEN,
CNRS, GREYC,
Caen, France 14050
Jalal.Fadili@ensicaen.fr

**Laurent Jacques**
ISPGroup/ICTEAM, FNRS
Université catholique de Louvain
Louvain-la-Neuve, Belgium 1348
laurent.jacques@uclouvain.be

## 1 Proof of Theorem 1 for $\alpha = 1$

As the proof presented in the paper for $\alpha = \infty$, our proof consists in construction a vector supported on $J$, obeying the implicit relationship (6) which becomes in this case

$$x_{\tau,J} = x_{0,J} + \widetilde{\Phi}^{-1}\Theta w - \tau v_{\infty,J}, \tag{12}$$

and which is indeed a solution to $(\mathcal{P}_{\infty}^{\tau}(\Phi x_0 + w))$ for an appropriate regime of the parameters $(\tau, \|w\|_{\alpha})$. Note that we assume that (INJ$_1$) holds and in particular, $\widetilde{\Phi}$ is invertible. If we define $Z_p \overset{\text{def.}}{=} \text{sat}(p)$, note that

$$\partial\|p\|_{\infty} = \left\{ u \ \middle| \ u_{Z_p^c} = 0, \langle u_{Z_p}, \text{sign}(p_{Z_p}) \rangle = 1, \text{sign}(u_{Z_p}) = \text{sign}(p_{Z_p}) \right\},$$

so that for an optimal primal-dual pair $(x, p)$, the condition (8) reads,

$$y_{Z_p^c} = \Phi_{Z_p^c, \cdot} x \ , \quad \langle \Phi_{Z_p, \cdot}^* \text{sign}(p_{Z_p}), x \rangle = \langle \text{sign}(p_{Z_p}), y_{Z_p} \rangle - \tau, \tag{13}$$

and

$$\text{sign}(y_{Z_p} - \Phi_{Z_p} x) = \text{sign}(p_{Z_p}). \tag{14}$$

To simplify the notations, we use

$$\Theta_p \overset{\text{def.}}{=} \begin{bmatrix} \text{Id}_{Z_p^c, \cdot} \\ \text{sign}(p_{Z_p}^*)\text{Id}_{Z_p, \cdot} \end{bmatrix} \quad \text{and} \quad \widetilde{\Phi}_p \overset{\text{def.}}{=} \Theta_p \Phi_{\cdot, J}.$$

One should then look for a candidate primal-dual pair $(\hat{x}, \hat{p})$ such that $\text{supp}(\hat{x}) = J$ and satisfying

$$\widetilde{\Phi}_{\hat{p}}\hat{x}_J = \tilde{y}_{\hat{p}} = \Theta_{\hat{p}} y - \tau\delta_{|J|} = \widetilde{\Phi}_{\hat{p}} x_{0,J} + \Theta_{\hat{p}} w - \tau\delta_{|J|}. \tag{15}$$

We now need to show that the first order conditions (7) and (8) hold for some $p = \hat{p}$ solution of the "perturbed" dual problem $(\mathcal{D}_{\infty}^{\tau}(\Phi x_0 + w))$ with $x = \hat{x}$. Actually, we will show that under the conditions of the theorem, this holds for $\hat{p} = p_{\infty}$, *i.e.*, $p_{\infty}$ is solution of $(\mathcal{D}_{\infty}^{\tau}(\Phi x_0 + w))$ so that

$$\hat{x}_J = x_{0,J} + \widetilde{\Phi}^{-1}\Theta w - \tau\widetilde{\Phi}^{-1}\delta_{|J|} = x_{0,J} + \widetilde{\Phi}^{-1}\Theta w - \tau v_{\infty,J}.$$

We remind that as defined in Section 1.4, $\Theta = \Theta_{p_\infty}$ and $\widetilde{\Phi} = \widetilde{\Phi}_{p_\infty}$ and that $v_{\infty,J} = \widetilde{\Phi}^{-1}\delta_{|J|}$. Let us start by proving the equality part of (7), $\Phi^*_{\cdot,J}p_\infty = \text{sign}(\hat{x}_J)$. Noting $\text{Id}_{I,J}$ the restriction from $J$ to $I$, we have

$$\text{sign}\left(x_{0,I} + \text{Id}_{I,J}\widetilde{\Phi}^{-1}\Theta w - \tau v_{\infty,I}\right) = \text{sign}\left(x_{0,I}\right)$$

as soon as

$$\left|\left(\widetilde{\Phi}^{-1}\Theta w\right)_i - \tau v_{\infty,i}\right| < |x_{0,I}| \quad \forall i \in I.$$

It is sufficient to require

$$\|\text{Id}_{I,J}\widetilde{\Phi}^{-1}\Theta w - \tau v_{\infty,I}\|_\infty < \underline{x}$$

$$\|\widetilde{\Phi}^{-1}\Theta\|_{\infty,\infty}\|w\|_\infty + \tau\|v_{\infty,I}\|_\infty < \underline{x},$$

with $\underline{x} = \min_{i\in I}|x_{0,I}|$. Injecting the fact that $\|w\|_\infty < c_1\tau$ (the value of $c_1$ will be derived later), we get the condition

$$\tau\left(bc_1 + \nu\right) \leqslant \underline{x},$$

with $b = \|\widetilde{\Phi}^{-1}\Theta\|_{\infty,\infty}$ and $\nu = \|v_\infty\|_\infty \leqslant b$. Rearranging the terms, we obtain

$$\tau \leqslant \frac{\underline{x}}{bc_1 + \nu} = c_2\underline{x},$$

which guarantees $\text{sign}(\hat{x}_I) = \text{sign}(x_{0,I})$. Outside $I$, defining $\text{Id}_{\tilde{J},J}$ as the restriction from $J$ to $\tilde{J}$, we must have

$$\Phi^*_{\cdot,\tilde{J}}p_\infty = \text{sign}\left(\text{Id}_{\tilde{J},J}\widetilde{\Phi}^{-1}\Theta w - \tau v_{\infty,\tilde{J}}\right).$$

From Lemma 1, we know that $-\text{sign}(v_{\infty,\tilde{J}}) = \Phi^*_{\cdot,\tilde{J}}p_\infty$, so that the condition is satisfied as soon as

$$\left|\left(\widetilde{\Phi}^{-1}\Theta w\right)_j\right| < \tau|v_{\infty,j}| \quad \forall j \in \tilde{J}.$$

Noting $\underline{v} = \min_{j\in\tilde{J}}|v_{\infty,j}|$, we get the sufficient condition for (7),

$$\|\widetilde{\Phi}^{-1}\Theta w\|_\infty < \tau\underline{v},$$

$$\|w\|_\infty < \tau\frac{\underline{v}}{b}. \tag{$c_1$a}$$

We can now verify (13) and (14). From (15) we see that (13) is satisfied i.e.,

$$y_{Z^c} = \Phi_{Z^c,J}\hat{x}_J \quad \text{and} \quad \text{sign}(p_{\infty,Z})^*\Phi_{Z,J}\hat{x}_J = \text{sign}(p_{\infty,Z})^*y_Z - \tau.$$

On $Z$, we have

$$y_Z - \Phi_{Z,\cdot}\hat{x} = w_Z - \Phi_{Z,J}\widetilde{\Phi}^{-1}\Theta w + \tau\Phi_{Z,J}v_{\infty,J}.$$

We know from Lemma 1 that

$$\text{sign}(p_{\infty,Z}) = \text{sign}\left(\Phi_{Z,J}v_{\infty,J}\right)$$

so that (14) holds, i.e., $\text{sign}(y - \Phi\hat{x})_Z = \text{sign}(p_{\infty,Z})$ as soon as

$$\|w_Z - \Phi_{Z,J}\widetilde{\Phi}^{-1}\Theta w\|_\infty \leqslant \tau\min_{i\in Z}|\Phi_{i,J}v_{\infty,J}|$$

$$\|\text{Id}_{Z,\cdot} - \Phi_{Z,J}\widetilde{\Phi}^{-1}\Theta\|_{1,\infty}\|w\|_1 \leqslant \underline{z}\tau$$

with $\underline{z} \stackrel{\text{def.}}{=} \min_{i\in Z}|\Phi_{i,J}v_{\infty,J}|$ and finally, noting $a \stackrel{\text{def.}}{=} \|\text{Id}_{Z,\cdot} - \Phi_{Z,J}\widetilde{\Phi}^{-1}\Theta\|_{1,\infty}$,

$$\|w\|_1 \leqslant \frac{\underline{z}}{a}\tau. \tag{$c_1$b}$$

($c_1$a) and ($c_1$b) together give the value of $c_1$. This ensures that $\hat{x}$ is solution to $(\mathcal{P}_1^\tau(\Phi x_0 + w))$ and $p_\infty$ solution to $(\mathcal{D}_\infty^\tau(\Phi x_0 + w))$, which concludes the proof. $\qquad\square$