[Reviews · NeurIPS 2016]

Reviewer 1

Summary

This paper analyzes solutions to the optimization problem min. ||x||_1 s.t. ||\Phi x - y|| \le \tau where the second norm is either the L1 or L-infinity norm. The authors give some dual characterization of the solution sets, proving that recovery of the optimal x's structure is sometimes feasible.

Qualitative Assessment

Given the crowded space in which this paper falls---there have been 100s (1000s?) of papers on sparse recovery in linear systems---it is imperative to carefully situate the paper and justify it in the context of NIPS. Somehow, I do not feel that this paper has really done that. The proof techniques are claimed to be novel--that's fine, but at some level they amount to constructing a dual vector that optimizes the problem under some noise conditions on w (where y = \Phi x + w), arguing that sign recovery/structure recovery is then possible. Of course, the dual witness technique has a long history in analysis of convex relaxations of sparsity problems, dating to Donoho, Candes, Tao, and Wainwright, so it is not clear that this really is particularly new. The experimental evaluation is a bit lacking as well. What is the experiment actually being run? Is there noise in the observation of y? What noise distribution on w is being used? My understanding of the justification for this paper was that variant noise distributions for w would change the analysis slightly (fine). But the experiments do not seem to study this... Moreover, Figure 2 seems to suggest that the classical L2 loss techniques are the best. Is this a consequence of the noise distribution used in the experiments? Is it because using L1 or L-infinity is actually not a good idea? I am not sure what to take home from the experiments; they seem to provide little explanatory evidence either way.

Confidence in this Review

2-Confident (read it all; understood it all reasonably well)


Reviewer 2

Summary

This paper analyzes support recovery of sparse vectors using an L1 regularizer and non-smooth fidelity terms like L0 or sup. They analyze support stability in terms of the dual certificate to small amounts of noise in these models.

Qualitative Assessment

To my best knowledge this is the first rigorous analysis of support stability using L0 and L\infty losses. I believe the contribution therefore is significant, although the main result is presented in terms of dual certificates which in my opinion diminishes the clarity. For example, when is the injectivity condition satisfied? Is this obvious in general? In addition, it is a bit hard to deduce from the main result how close the estimated support J is to the actual support I. In other words, for a given noise level and even perhaps given vector magnitudes, how much larger can J be than I? The experiments seem to attempt to address this question more clearly, but again I would also like to see how (support of) the actual minimizer x_\tau compares with (the support of) the true signal. I think a reformulation of the main result and experiments might make for much clearer presentation. In summary, I find the contribution to be strong but that it is presented in a manner that makes the parameters of interest obscure. Either the authors should comment why the extended support itself is of interest, or rephrase the results to address the actual recovered support.

Confidence in this Review

2-Confident (read it all; understood it all reasonably well)


Reviewer 3

Summary

The authors present sharp criteria for l_1-sparse support recovery under small noise in the context of non-smooth regression losses l_\infty and l_1. Traditionally, the problem studied in support recovery left the loss shape fixed at MSE (l2^2) loss and explored different non-smooth penalties, starting from the work of Fuchs on l1 penalty up until the recent, possibly most general partly smooth penalties of Vaiter et al. In the present contribution, the authors keep the simple l1 penalty but explore the effect of changing the loss function. Concretely, two loss functions, l_\infty and l_1 are studied. Both share the characteristic of being polyhedral. A proof for both cases is given, along with some confirmatory experimental results.

Qualitative Assessment

This paper presents a new angle of generalization of the l_1 support recovery criteria to other types of loss functions, which are generally useful in regression: l_\infty corresponds to a uniform noise assumption, whereas l_1 corresponds to a sparse noise assumption. In performing this generalization, it exhibits the functionality of some quantities also obtained in the classic MSE + l1 setting, but which at the time were too specific to be given names. We thus have a non-trivial step forward in the understanding of the mechanisms involved. The proofs in the main paper are sound. However, my main comments concern the readability of this contribution. This paper took me by far the longest to review, because while mathematically perfectly rigorous, it is as if the authors a) underestimate the fact that their contribution is some non-trivial convex analysis, that they b) do not seem to be interested in making any concessions to the reader by providing their own intuitions or figures elucidating them and c) that this may lead to reviews that completely miss the point of the paper. Specifically, my points are * It would be great to have a small diagram explaining the regression problem in 1D or 2D, especially since l_1 and l_\infty losses are equal, resp. equivalent in these two settings. There is sufficient space to add this without deleting anything. * up until a certain point in the paper, the paper works with general dual pairs \alpha, \beta whose harmonic mean is 1. At some point this specializes to the polyhedral losses and later in the paper specializes to the l_\infy loss (l1 in supp mat). These transitions have to be made more clear. It would be helpful also to state what intuitions are needed to keep the general (\alpha, \beta) (i.e. \alpha \in (1, \infty) is not polyhedral so the proof is different, but here is how you would go about it, but here is why we don't do this in this paper). * In general, the first reading of this paper is difficult, because many notations are introduced a posteriori. One arrives at table 1 and spends some time looking for definitions of \tilde\Phi and S. These are introduced in the lines below. Then one assumes |S| = |J|, which seems surprising, and is referred forward to Lemma 2 for discussion, instead of giving a quick intuition that this is not only "for simplicity" but even "almost surely holds" in a probabilistic setting and is a reasonable assumption, before referring to the details mentioned in lemma 2. There are also several formulae which introduce a symbol and state in the next lines ", where [symbol] means [meaning]". Could these be put before the formula? * l120 concerning the situation where I != J, there is also [Tibshirani, The Lasso problem and uniqueness]. While the paper seems to state something that is implicit in many previous contributions, it would be worth being a little more explicit in how these relate * It took me 3 readings to see the bold r in regularization in l139 to make the link to FO_r. Please introduce explicitly FO_r and FO_\beta. Additionally, it would be helpful to make it a bit clearer, by adding an intermediate step, what "corresponds" means in l139. Not all readers are versed enough in convex duality to read this easily. In general, since there still remains half a page of space, a way of obtaining the Fenchel dual (e.g. by introducing the lagrangian, splitting primal variables and minimizing it wrt both of them) and first order conditions could also be given. * It would be great to have a bit more intuition on the Lagrange multipliers v_\beta (and some more intuition on p_\beta can be helpful too). In which space do they live (the same as x resp. same as y). How can one imagine them geometrically? The polytope defined by the regression constraint is difficult to untangle in the space of x, since one doesn't know which constraints are active. This polytope is an intersection of seminorm balls, which could even be illustrated in 2D. Diagrams, although not easy to make, would help readers enormously. * between line 189 and 190, does it have to be p_{1, S}? Minor * l192 "So now, " -> "Now " * l194 hypotheses -> hypothesis * between l200 and l201 put a word between the formulas "or the second line which implies the first by operator inequality" *l202 get -> obtain *l207 get -> obtain *l248 exceeds -> exceed *l255 theoertical -> theoretical

Confidence in this Review

2-Confident (read it all; understood it all reasonably well)


Reviewer 4

Summary

This papers provides support recovery guarantees of sparse regression problem when data term is a l_p norm with p = {1,2+\infty}. The guarantees are derived from the constrained formulation.

Qualitative Assessment

This paper arises a very interesting question. It is dealt seriously with rigorous theoretical developments and adapted experiments.

Confidence in this Review

2-Confident (read it all; understood it all reasonably well)


Reviewer 5

Summary

This paper studies the support recovery guarantess of the variants of the relaxed basis pursuit algorithm. The loss studied here is l1 loss and infinity loss instead of the traditional l2 loss. This work extends the existing theory for l2 loss case. The main theorem gives guarantess for support recovery under a specific small noise level setting.

Qualitative Assessment

The main contribution of this paper is to give detailed support recovery guarantees for l1 and infinity loss variant of relaxed basis pursuit. This completes the theorectal analysis of this problem in these specific cases. The paper explains the main problem setup, main results and experiments very clearly. There is a minor typo like - line 62 "explicitely" Here are some main weak points of the paper. - The use of l1 loss and infinity loss is motivated by different noise models. However, in simulations it is not clear which noise model is used. It would be interesting to see how three methods perform differently under different noise models. Simulations basically show l2 loss is superior for support recovery, if no practically interesting cases are provided, why are the theoretical support recovery guarantees interesting? - Theorem 1 only applies for a small noise setting. It does not provide insights when the noise level is larger than what is assumed to be. Like it is discussed in line 215, it could that the Theorem 1 only applies to the case where no nosie is allowed. Could the authors elaborate more on large nose settings, either in theory or in simulations? - Most important information about the numerical experiments is given, however the noise level is not specified in any experiments.

Confidence in this Review

2-Confident (read it all; understood it all reasonably well)


Reviewer 6

Summary

Given noisy linear observations of an underlying model, authors considered an \ell_1 norm minimization problem subject to data fidelity constraints given as \ell_\alpha norms of the error. Support recovery guarantees for such an optimization problem are given. While in some parts the proofs are presented for special cases of \ell_1 and \ell_\infty constraints, the authors provide insights on results for any \ell_\alpha constraint for \alpha\geq 1.

Qualitative Assessment

The presentation of the paper is excellent. Figure 2 provides a nice intuition into the behavior of \ell_alpha constraints. A suggestion: it would be interesting to see the plots of Figure 2 when 1/alpha is plotted against k. I suspect we will then observe a symmetric behavior around alpha=2. 1/alpha seems like a natural choice for understanding the behavior of \ell_alpha norms. In the numerical experiments, it would be nice to comment on how the choice of x_0 (equal nonzero values in absolute value) affects the results. Minor typos: - line 118: "is often coined" to "is coined as" or "is often referred to as" - it would be nice to describe the names of the two equations (FO) in the beginning of section 2 inside the text. - line 139: "r" should not be in boldface. - line 186: first sentence can be reworded. - The spacing for Figure captions can be improved.

Confidence in this Review

2-Confident (read it all; understood it all reasonably well)